

# Land abandonment as driver of woody vegetation dynamics in Tamaulipan thornscrub at Northeastern Mexico

Eduardo Alanís-Rodríguez[1], Cristian A. Martínez-Adriano[1], Laura Sanchez-Castillo[2], Ernesto Alonso Rubio-Camacho[3] and Alejandro Valdecantos[4]

[1] Faculty of Forestry, Autonomous University of Nuevo Leon, Linares, Nuevo Leon, Mexico
[2] Faculty of Engineering and Science, Autonomous University of Tamaulipas, Ciudad Victoria, Tamaulipas, Mexico
[3] Experimental Field Centro Altos de Jalisco, National Institute of Research for Forestry, Agricultural and Livestock, Tepatitlan de Morelos, Jalisco, Mexico
[4] Department of Ecology, University of Alicante, San Vicente del Raspeig, Alicante, Spain

Corresponding author
Cristian A. Martínez-Adriano,
cristian.martinez.cama@gmail.com

## ABSTRACT

**Background:** Vegetation structure is defined as the temporal and spatial distribution of plant species in a particular site. Vegetation structure includes vertical and horizontal distribution and has been widely used as an indicator of successional changes. Ecological succession plays an essential role in the determination of the mechanisms that structure plant communities under anthropogenic disturbances. After an anthropogenic disturbance, such as grazing, forests follow changes in the original composition and vegetation structure, which eventually could restore some of their attributes to become mature forests again. To know how the time of abandonment affects woody plant communities, we ask the following questions: (1) How does the species richness, diversity, and vertical structure (A index) change concerning the time of abandonment? (2) Are species similarities among woody vegetation communities determined by land abandonment? (3) Which woody species have the highest ecological importance in each successional stage?

**Methods:** We explored how successional stages after land abandonment mediated the species richness, species diversity (alpha and beta), and ecological importance value index on four areas of Tamaulipan thornscrub. We selected four areas that differed in time of abandonment: 10, 20, 30, and >30 years. The first three areas were used for cattle grazing, whereas the >30-year area was selected as a control since it does not have a record of disturbance by cattle grazing or agriculture. During the summer of 2012, we randomly established four square plots (40 m × 40 m) in each area, separated at least 200 m from each other. In each plot, we recorded all woody individuals per species with a basal diameter ≥1 cm at 10 cm above ground level. We estimated species richness indices, species diversity (alpha and beta), and ecological importance value index.

**Results:** We recorded 27 woody species belonging to 23 genera and 15 families. Fabaceae accounted for 40% of the species. *Acacia farnesiana* was the most important and abundant species in the first three successional stages. We suggested that older successional stages of Tamaulipan thornscrub promote woody plant communities, characterized by a higher complex structure than younger communities. We observed the highest species similarity between the sites with a closer time of

abandonment, while the lowest similarity was shown between the sites with extreme time of abandonment. We conclude that Tamaulipan thornscrub shows a similar trend of ecological succession to other dry forests and the time of abandonment has a high mediation on plant dynamics in the Tamaulipan thornscrub. Also, we stand out the importance of secondary forests for Tamaulipan thornscrub woody plant communities. Finally, we recommended future studies include aspects of regeneration speed, the proximity of mature vegetation, and the interactions of plants with their seed dispersers.

## INTRODUCTION

Tropical dry forests are among the most threatened ecosystems in Latin America (*García-Millán & Sanchez-Azofeifa, 2018*), due to the land-use change for many productive activities among we can include agriculture and cattle expansion (*Portillo-Quintero & Sánchez-Azofeifa, 2010*). These activities affect directly the dry forest biodiversity and structure since these ecosystems have one of the most endemisms in the globe (*Sánchez-Azofeifa et al., 2009*).

Vegetation structure is defined as the temporal and spatial distribution of plant species in a particular site; it has been widely used as an indicator of successional changes through time after a process of disturbance in natural ecosystems (*Domínguez-Gómez et al., 2013*; *Kulmatiski & Beard, 2019*). Vegetation structure includes vertical and horizontal distribution of vegetation, spatial patterns, and size and age of trees (*Oliver & Larson, 1996*). These factors are important in ecosystem studies since it can be inferred whether an ecosystem reached the mature successional stage (*Domínguez-Gómez et al., 2013*). Studies of ecological succession play an important role in the determination of the relative importance of mechanisms that structure plant communities under different environmental conditions (*Choung et al., 2020*; *McCook, 1994*; *van Andel, Bakker & Grootjans, 1993*). Additionally, understanding the forest successional dynamics in forests is a crucial issue for the establishment and design of conservation strategies (*Sánchez-Azofeifa et al., 2005*; *Quesada et al., 2009*). According to *Falkowski, Chankin & Diemont (2020)*, several factors affect secondary forest succession, among them: abiotic conditions, the surrounding space, and land-use history (*Quesada et al., 2009*). The latter has been recognized as one of main drivers of changes in vegetation successional dynamics (*Baeza et al., 2007*; *Quesada et al., 2009*; *Pequeño-Ledezma et al., 2018*). The land-use change is defined as the modification of native vegetation into areas for productive purposes (*Alanís-Rodríguez et al., 2015a*; *Falkowski, Chankin & Diemont, 2020*; *Pequeño-Ledezma et al., 2018*; *Zhang et al., 2018*) like cattle grazing, agriculture, or mineral extraction.

After an anthropogenic disturbance, such as logging, grazing, or farming, forests follow a revegetation process characterized by changes in the original floristic composition and vegetation structure (*Kitenberga et al., 2020*; *Lohbeck et al., 2020*; *Tálamo et al., 2020*;

*Zhang et al., 2018*). These natural successional processes eventually could restore the attributes of degraded ecosystems in old-growth forests (*DeWalt, Maliakal & Denslow, 2003*). Nevertheless, despite their importance, the successional trajectories of plant communities (mainly in dry or deciduous forests) after land abandonment are still poorly understood (*Bonet & Pausas, 2004*; *Quesada et al., 2009*). Thus, the changes in vegetation structure, composition, and abundance are important to understand the potential driver of long-term land abandonment on the configuration of mature ecosystems of Tamaulipan thornscrub.

*Sarmiento-Muñoz et al. (2019)* found that the Tamaulipan thornscrub areas used for grazing could affect the natural ecological succession of the ecosystem. Recently, several studies highlighted the structure of regenerated woody communities resulting from secondary succession (*Alanís-Rodríguez et al., 2008*; *González-Rodríguez et al., 2010*; *Jiménez-Pérez et al., 2009*, *2012*, *2013*; *Patiño-Flores et al., 2021*), but none of them explored the successional trajectories after land abandonment under a well-defined chronosequence. Although plant diversity has been considered in many studies on Tamaulipan thornscrub and its importance is well known (*Alanís-Rodríguez et al., 2015a*). It is not yet clear how land abandonment (successional changes) contributes to the changes in woody plant diversity, how the tree and shrub species are replaced by others, or how these replacement contributes to the general structure of Tamaulipan thornscrub.

In this study, we explored how the successional stages of four areas of Tamaulipan thornscrub drive the changes in species richness, species diversity (alpha and beta), vertical structure of vegetation, and ecological importance value index. Specifically, we wanted to know: (1) How does species richness, diversity, and vertical structure (A index) change concerning the time of abandonment? (2) Are species similarities among vegetation communities related to land-use abandonment? (3) Which species are the most important within each successional stage? According to this background and questions, we elaborated three main hypotheses: (1) the earlier successional areas would have lower values of species richness, species diversity, and the species height profile index than sites with the longest time of abandonment; (2) the species similarity among sites will be related to the time of abandonment; therefore we would find the lowest species similarity between the sites with early and late years of abandonment; and (3) the species with a higher ecological importance value in early successional stages will not have high importance values in sites with vegetation on late-successional stage.

## MATERIALS AND METHODS

### Study area

The study area was located in a Tamaulipan thornscrub in Linares, Nuevo Leon state, Northeastern Mexico (25°09′–24°33′N and 99° 54′–99°07′W; Fig. 1) at an elevation of 350–411 m a.s.l. This area has an arid desert-warm climate with a maximum amount of rainfall during September (*INEGI, 1986*). The area has an annual average temperature of 23.7 °C and annual cumulative precipitation of 810 mm (*Martínez-Adriano et al., 2021*). Low hills with gentle slopes are predominant in the area, mainly with Vertisol soils (*INEGI,*

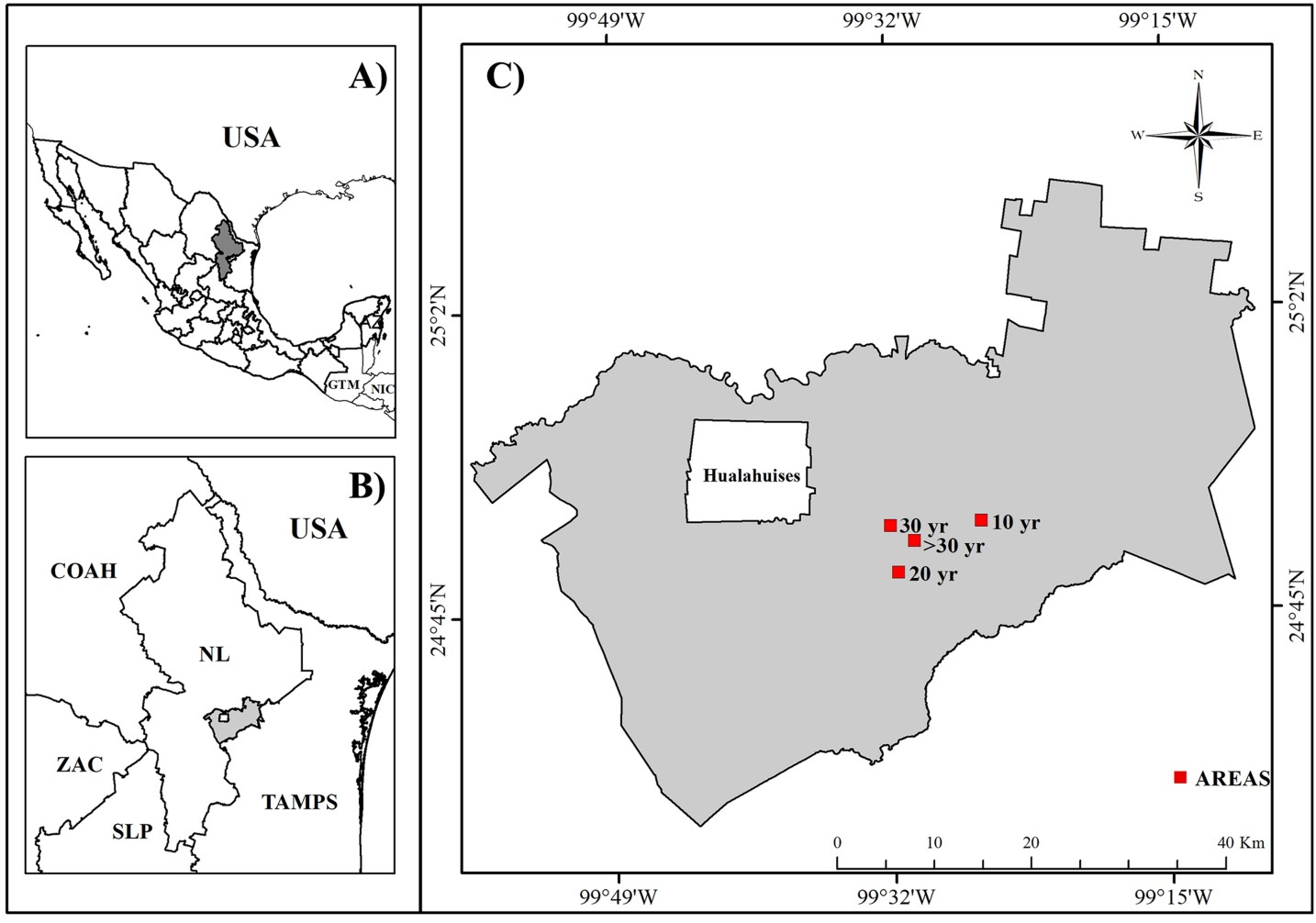

**Figure 1 Study area location.** (A) Mexico state division. (B) Nuevo Leon State municipalities. (C) Studied areas localization within Linares municipality. Source credit: Nelly A. Leal-Elizondo.

*1986*). The four selected areas had similar topographic conditions and only differed in the years of abandonment.

We selected four areas with different time of abandonment after last recorded human disturbance occurs (10, 20, 30, and >30 years). The first three areas were used for cattle grazing, whereas the >30-year area was selected as a control. The native vegetation in the three grazed areas was removed with heavy machinery and replaced with forage grass (*Cenchrus ciliaris* L.) as food for cattle. Grazing areas were excluded 10, 20, 30, and >30 years before the start of this study and it allowed the natural plant succession. The evaluated areas were located next to plant communities that are growing after livestock activity. The control area (>30 yr) shows Tamaulipan thornscrub vegetation without evidence of disturbance or land-use changes. Local people used the control area exclusively to collect wild fruits and firewood from the ground. This information was corroborated by satellite images and personal interviews. Additionally, we highlighted that

EA obtained verbal permission from the owners of the four sites used to conduct this study.

## Experimental design

During summer of 2012, we randomly established four square plots (40 m × 40 m) in each condition of abandonment and separated by a minimum distance of 200 m. In each plot, we recorded all woody individuals per species with a basal diameter ≥1 cm at 10 cm above ground level. Total height (h) and basal diameter ($d_{0,10}$) were measured in all individuals of woody species.

We collected botanical samples of the woody plant species found in each plot for taxonomic identification. The taxonomical identification was carried out by botanical experts and by comparing our samples with specimens stored in the herbarium at Facultad de Ciencias Forestales at Universidad Autónoma de Nuevo León. To confirm the current scientific names, we used The Plant List web database (*The Plant Life, 2013*).

## Species richness and diversity

We determined the species richness as the number of species in each plot. Additionally, with the identity and abundance of each woody species, we calculated Shannon's diversity index ($H' = -\sum_{i=1}^{S} p_i log_e p_i$), where $S$ is the number of species, and $pi$ is the proportion of individuals of species $i$ concerning the total number of individuals (*Shannon, 1948*). With H´ values per site, we estimated the effective number of species ($^1D$), based on the exponential of Shannon's index (exp ($H'$)) (*Cultid-Medina & Escobar, 2019*; *Jost, 2006*; *Moreno et al., 2011*). Additionally, we calculated the vertical distribution of the species through the species height profile index (A-index), which is also based on Shannon's index ($H'$). The A-index takes classifies the vertical distribution of species into three height zones, zone I goes from 100–80% (where the tallest tree is 100%), zone II (80–50%), and zone III (50–0%) (see *Pretzsch (2009)*). Additionally, to know if years of abandonment affect species similarity, we performed a cluster analysis of species similarity in PAST software ver. 3.24 (*Hammer, Harper & Ryan, 2001*). This analysis was performed with Bray-Curtis similarity and UPGMA algorithm and 9,999 permutations (this procedure allows to support each branch of the resultant grouping tree; *Hammer, Harper & Ryan, 2001*).

## Importance values index

The importance value index is composed of three elements (*Abundance, Dominance, and Frequency*) given by species within the studied community. We estimated the absolute abundance per species ($A_i$) through dividing the number of records of species I ($N_i$) by the sampling plot area (S, in m$^2$). We calculated the relative abundance of species using the equation: RA$_i$ = ($A_i$/ΣA$_i$) * 100, where RA$_i$ is the relative abundance of species i and ΣA$_i$ is the total abundance of species (*Mostacedo & Fredericksen, 2000*). To estimate dominance, we extrapolated the basal area per hectare (G ha$^{-1}$) for each species. Absolute dominance is given by D$_i$ = Ba$_i$/S, where D$_i$ is the absolute dominance of species i, Ba$_i$ is the basal area of species i, and S is the area of the sampling plot. Relative dominance was calculated using

the equation $Rd_i = (D_i/\Sigma D_i) * 100$, where $Rd_i$ is the relative dominance of species i, and $\Sigma D_i$ is the total dominance of species. Finally, the relative frequency is $Rf_i = (F_i/\Sigma F_i) * 100$, where $Fr_i$ is the relative frequency of species i, $F_i$ is the number of sites where the species i is present and $\Sigma F_i$ is the total frequency of all species (*Edwards, May & Webb, 1994*; *Mostacedo & Fredericksen, 2000*). Thus the importance value index (IVI) per species (i) was computed as follows: $IVI_i = (Ar_i + Dr_i + Fr_i)/3$ (*Mueller-Dombois & Ellenberg, 1974*).

The volume above the ground of each site ($m^3ha^{-1}$) was calculated as a measure of woody plants per area. The volume per individual was estimated with the formula: $V = g * h * MC$. Where V is the volume, g is the assumed circular section of the stem diameter ($d_{0.10}$), h is the total height of the individual, and MC is the morphic coefficient. The value of MC for the Tamaulipan thornscrub is a constant of 0.5 (*Alanís-Rodríguez, Mora-Olivo & Marroquín de la Fuente, 2020*).

### Effect of years of abandonment on vegetation structure

To evaluate the differences among areas we used the analysis of variance and HSD posthoc Tukey test when differences were found (both functions from "agricolae" R package; *de Mendiburu, 2020*). ANOVA assumptions of normality and homoscedasticity were tested graphically through residual plots and analytically with functions Shapiro test (for normality) and Levene test (for homoscedasticity), both functions inside "car" R package (*Fox & Weisberg, 2019*). When the assumption of equal variances was not accomplished, we used the ANOVA test assuming unequal variances (on the one-way function) and the Games-Howell posthoc test from "userfriendlyscience" package (*Peters, 2018*) for the R software ver. 4.0.2 (*R Core Team, 2020*).

## RESULTS

We recorded 27 woody species, belonging to 23 genera and 15 families. The families with highest species number were Fabaceae (11 species), Rubiaceae (two), and Rutaceae (two). While the most representative genera were *Acacia* (three), *Parkinsonia*, and *Randia* (both with two species each; see Table 1). We observed *Acacia farnesiana* as the unique plant species found in all conditions of abandonment, while species like *Cordia boissieri*, *Celtis pallida*, *Prosopis laevigata*, and *Sideroxylon celastrinum* were observed in areas ≥20 yr of abandonment. On the other hand, we found that *Bernardia myricifolia*, *Condalia hookeri*, *Eysenhardtia polystachya*, *Leucophyllum frutescens*, and *Yucca filifera* only occurring in >30 yr site (Table 1).

Species richness and effective number of species. We found that the >30 yr site showed the highest species richness (21 taxa) and the number of species decreased as the disturbance became more recent ($F_{3, 12} = 17.43$; $p < 0.001$). Likewise, for the effective number of species ($^1D$), we found a similar pattern ($F_{3, 12} = 22.76$; $p < 0.001$), where the >30 yr site had the highest number of effective species ($^1D = 9.3$), followed by the 30 yr ($^1D = 4.53$), 20 yr ($^1D = 2.39$), and the lowest was found in the 10 yr site ($^1D = 1.28$).

Species Height Profile Index (A-index). For this index, the residuals were normally distributed ($W = 0.98$, $p = 0.97$) and the variance was not homogeneous (Levene's test:

**Table 1 List of woody plant species recorded in the study area.**

| Family | Plant species |
| --- | --- |
| Boraginaceae | *Cordia boissieri* A. DC. |
| Cannabaceae | *Celtis pallida* Torr. |
| Koeberliniaceae | *Koeberlinia spinosa* Zucc. |
| Ebenaceae | *Diospyros texana* Scheele |
| Euphorbiaceae | *Bernardia myricifolia* (Scheele) S. Watson |
| Fabaceae | *Acacia amentacea* DC. |
| Fabaceae | *Acacia farnesiana* (L) Willd. |
| Fabaceae | *Acacia rigidula* Benth. |
| Fabaceae | *Caesalpinia mexicana* A. Gray |
| Fabaceae | *Ebenopsis ebano* (Berland.) Barneby & J. W. Grimes |
| Fabaceae | *Eysenhardtia polystachya* (Ortega) Sarg. |
| Fabaceae | *Havardia pallens* (Benth.) Britton & Rose |
| Fabaceae | *Mimosa monancistra* Benth. |
| Fabaceae | *Parkinsonia aculeata* L. |
| Fabaceae | *Cercidium texanum* A. Gray |
| Fabaceae | *Prosopis laevigata* (Willd.) M. C. Johnst. |
| Asparagaceae | *Yucca filifera* Chabaud |
| Oleaceae | *Forestiera angustifolia* Torr. |
| Rhamnaceae | *Condalia hookeri* M. C. Johnst. |
| Rubiaceae | *Randia obcordata* S. Watson |
| Rutaceae | *Helietta parvifolia* (A. Gray) Benth. |
| Rutaceae | *Zanthoxylum fagara* (L.) Sarg. |
| Sapotaceae | *Sideroxylon celastrinum* (Kunth) T. D. Penn. |
| Scrophulariaceae | *Leucophyllum frutescens* (Berland.) I. M. Johnst. |
| Simaroubaceae | *Castela erecta* Turpin |
| Zygophyllaceae | *Porlieria angustifolia* (Engelm.) A. Gray |

**Note:**
The scientific names of woody plants were obtained from The Plant List web page (TLP 2013).

F = 4.58; $p$ = 0.02). The one-way analysis showed differences among the areas with times of abandonment ($F_{3, 6.25}$ = 21.135; $p$ = 0.0011). The highest altitudinal floors were recorded in the >30 yr area (A = 2.68 ± 0 .18), while the lowest altitudinal floors were observed in the 10 yr area (A = 1.01 ± 0.37).

## Importance value index

Abundance. We found that the area >30 yr showed the highest abundance of woody plants (1,828 ± 299 N·ha$^{-1}$; $F_{3, 12}$ = 4.824; $p$ = 0.019), while the lowest abundance was observed in the 30 yr area (762 ± 336 N ha$^{-1}$) (Table 2). We observed that *Acacia farnesiana* was the most abundant woody species in all successional stages except in the >30 yr area. Nevertheless, the abundance of *A. farnesiana* decreased as the time of abandonment increased (from 1,426 to 156 individuals ha$^{-1}$) and became less abundant than *Havardia*

**Table 2 Woody plant species recorded in the study area.**

| Species | 10 years | | | 20 years | | | 30 years | | | >30 years | | |
|---|---|---|---|---|---|---|---|---|---|---|---|---|
| | Nha | Gha | IVI | Nha | Gha | IVI | Nha | Gha | IVI | Nha | Gha | IVI |
| A. amentacea | 70.31 | 0.20 | 15.73 | | | | | | | 215.63 | 0.49 | 8.17 |
| A. farnesiana | **1,415.63** | **1.97** | **76.16** | **931.25** | **5.88** | **60.56** | **459.38** | **6.02** | **49.49** | 156.25 | 0.53 | 7.24 |
| A. rigidula | | | | | | | 7.81 | 0.05 | 1.64 | | | |
| B. myricifolia | | | | | | | | | | 1.56 | <0.01 | 0.64 |
| C. boissieri | | | | 20.31 | 0.19 | 4.66 | 4.69 | 0.02 | 2.53 | 185.94 | 1.77 | 12.45 |
| C. erecta | | | | | | | 26.56 | 0.13 | 5.03 | 4.69 | 0.01 | 0.72 |
| C. hookeri | | | | | | | | | | 51.56 | 0.38 | 4.76 |
| C. mexicana | | | | 23.44 | 0.13 | 2.87 | | | | | | |
| C. pallida | | | | 29.69 | 0.04 | 2.62 | 17.19 | 0.06 | 5.45 | 28.13 | 0.09 | 2.06 |
| D. texana | | | | | | | 45.31 | 0.14 | 5.90 | **429.69** | **2.34** | **19.01** |
| E. ebano | | | | 4.69 | 0.07 | 2.03 | | | | 31.25 | 0.08 | 1.47 |
| E. polystachya | | | | | | | | | | 54.69 | 0.19 | 4.08 |
| F. angustifolia | | | | 15.63 | 0.02 | 2.13 | | | | 4.69 | 0.03 | 0.78 |
| G. angustifolium | | | | 1.56 | 0.00 | 1.64 | | | | 6.25 | 0.02 | 1.99 |
| H. pallens | 9.38 | 0.02 | 4.16 | 28.13 | 0.13 | 4.58 | | | | 434.38 | 1.30 | 15.18 |
| H. parvifolia | | | | 3.13 | 0.01 | 1.71 | | | | 31.25 | 0.11 | 2.77 |
| K. spinosa | | | | | | | 9.38 | 0.02 | 1.60 | | | |
| L. frutescens | | | | | | | | | | 1.56 | 0.01 | 0.65 |
| M. monancistra | | | | 1.56 | 0.00 | 1.64 | | | | | | |
| P. aculeata | | | | | | | 20.31 | 0.18 | 2.74 | | | |
| P. laevigata | | | | 85.94 | 0.67 | 11.95 | 90.63 | 0.99 | 12.53 | 20.31 | 0.07 | 1.21 |
| P. texana | | | | | | | 31.25 | 0.24 | 4.59 | 31.25 | 0.18 | 3.05 |
| R. obcordata | | | | 4.69 | 0.00 | 1.73 | | | | 3.13 | 0.01 | 0.68 |
| S. celastrinum | | | | 7.81 | 0.02 | 1.88 | 28.13 | 0.09 | 3.83 | 78.13 | 0.51 | 5.72 |
| Y. filifera | | | | | | | | | | 4.69 | 0.55 | 3.35 |
| Z. fagara | 4.69 | 0.01 | 3.95 | | | | 21.88 | 0.09 | 4.66 | 53.13 | 0.18 | 4.04 |
| Totals | 1,500.00 | 2.20 | 100.00 | 1157.81 | 7.15 | 100.00 | 762.50 | 8.03 | 100.00 | 1,828.13 | 8.86 | 100.00 |

Note:
The absolute abundance (Nha), dominance (Gha), and importance value index (IVI) of each plant species per condition of abandonment. The highest values for each site is highlighted in bold.

pallens, *Diospyros texana*, *Acacia amentacea*, and *Cordia boissieri* in the >30 yr area (Table 2).

Dominance. The dominance showed differences among areas ($F_{3, 12}$ = 5.629; $p$ = 0.012). The highest values of dominance were recorded in the >30 yr site with 8.86 ± 1.60 G ha$^{-1}$, however, this site only showed differences from the 10 yr site (2.19 ± 0.52 G ha$^{-1}$; Table 2). We observed that *Acacia farnesiana* was the most dominant species in the three earliest successional stages. While, in the >30 yr area the species with highest dominance were *Diospyros texana*, *Cordia boissieri*, and *Havardia pallens* (Table 2).

Volume. We found that the control area (>30 yr) showed the highest wood volume (22.21 ± 3.61 m$^3$ ha$^{-1}$; $F_{3, 12}$ = 4.765; $p$ = 0.0206), and the lowest was observed in the 10 yr

**Table 3 Bray-Curtis similarity matrix.**

|  | 10 | 20 | 30 | >30 |
|---|---|---|---|---|
| **10** | 1.000 | **0.708** | 0.410 | 0.145 |
| **20** |  | 1.000 | **0.599** | 0.186 |
| **30** |  |  | 1.000 | 0.255 |
| **>30** |  |  |  | 1.000 |

**Note:**
The highest values of similarity between paired areas is shown in bold.

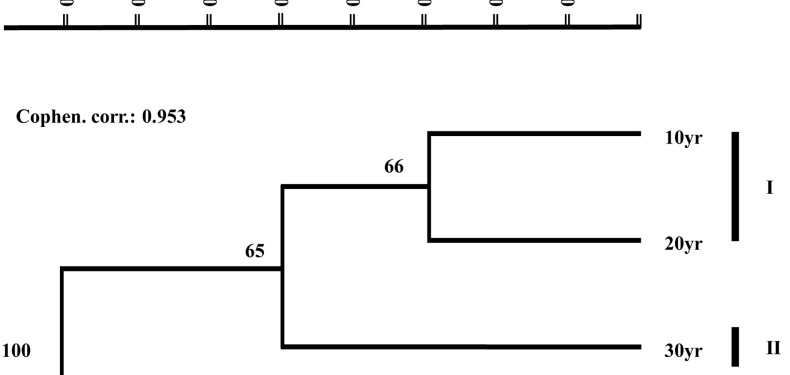

**Figure 2 Bray-Curtis similarity among species diversity of the studied communities.** The numbers close to each branch are the percentage of permutations that each node is still supported on the final grouping dendrogram (see *Hammer, Harper & Ryan (2001)*). The cophenetic correlation (Cophen. corr.) obtained, showed a good measure of degree of fit for the classification of our data set (*Saraçli, Doğan & Doğan, 2013*).

area ($4.13 \pm 0.48$ m$^3$ ha$^{-1}$). Wood volumes were $21.45 \pm 13.55$ for the 20 yr site and $19.58 \pm 6.94$ m$^3$ ha$^{-1}$ for the 30 yr site.

IVI. We observed that *Acacia farnesiana* was the most important species in all successional stages, except in the control area (>30 yr). Nevertheless, as the time after perturbation increased other species appear to be more important, especially those with tree-like characteristics (tall trees with wide canopy) such as *D. texana*, *C. boissieri*, and *H. pallens* (Table 2).

## Similarity among areas with different years of abandonment

We found the highest similarity between areas with closer years of abandonment and decreased with the temporal distance of abandonment. The most similar areas were 10 and 20 yr, followed by 20 and 30 yr (Table 3). The cluster analysis showed three well-defined groups (Cophenetic correlation = 0.95), one formed by 10 and 20 yr sites, and the 30 yr and >30 yr sites in solitary branch each (Table 3; Fig. 2).

## DISCUSSION

In this study, we simultaneously explored the changes in species richness, diversity, and importance value index among four sites with different times of abandonment in the Tamaulipan thornscrub.

We found 27 woody species in this study which represents the same number of plant species reported by *Jiménez-Pérez et al. (2009)* and similar to the species recorded by *Alanís-Rodríguez et al. (2013*, *2015b)* (30 plant species) in the Tamaulipan thornscrub. While *Canizales-Velázquez et al. (2009)* reported 52 plant species in a piedmont scrub (a value almost two times higher than our results). This means that our study represents at least more than half of the plant species of these previous studies. Additionally, we observed that Fabaceae was the family with the highest number of species. These results agree with previous floristic studies in remnants of Tamaulipan thornscrub, where Fabaceae was the family with a higher number of species (*Alanís-Rodríguez et al., 2015b*; *González-Rodríguez et al., 2010*; *Jiménez-Pérez et al., 2009*; *Molina-Guerra et al., 2013*). This indicates that the family possibly has species with the potential to adapt to the conditions that occur at sites with different successional stages (*e.g.*, humidity, exposure, radiation).

Our first hypothesis said that the early successional sites would have lower values of species richness, diversity, and A index than sites with earlier-time of abandonment. This prediction is supported by our results since our findings showed that species richness, effective species index, and A index were higher in the >30 yr site, while the number of species decreased when the successional stage was more recent. These results are similar to the general pattern for scrubs found in previous studies where the oldest plant communities have higher plant species richness and diversity (*Abella, 2010*; *Alanís-Rodríguez et al., 2018*; *Connell, 1978*; *Ugalde, Granados-Sánchez & Sánchez-González, 2008*). Similarly, several studies highlight an increase in species richness and diversity of tree communities from early to late successional stages in tropical deciduous forests (*Quesada et al., 2009*). However, previous studies found that certain attributes of plant communities (as basal area) are predicted adequately from chronosequences (*Chazdon et al., 2007*; *Pascarella, Aide & Zimmerman, 2004*) whereas differences are found concerning species richness and species composition (*Chazdon et al., 2007*; *Johnson & Miyanishi, 2008*; *Pascarella, Aide & Zimmerman, 2004*; *Sheil, 2001*). Regarding abundance and biomass (volume) we found similar patterns that a study on the Sonoran thornscrub, which also found that older-growth forests have a more complex structure, higher biomass, and individual densities than younger ones (*Álvarez-Yépiz et al., 2008*). Thus, we corroborated that plant communities that occur in later successional stages within Tamaulipan thornscrub, also could have more complex structure (in both horizontal and vertical stratification), than those that occur in early successional stages. This makes us notice the importance of ecological succession in the dynamics and structure of Tamaulipan thornscrub ecosystems.

Our second hypothesis said that species similarities among sites will be related to the time of abandonment; therefore, we would find the lowest species similarity between the

sites with early and late years of abandonment. Our results supported this prediction and were similar to the findings of *Alanís-Rodríguez et al. (2018)* since the highest similarity occurred among areas with a similar time of abandonment. Additionally, we found the same pattern of similarity to the findings reported by *Sánchez-Reyes et al. (2021)*, where the lowest similarity was observed between the two sites with an extreme time of abandonment (10 and 30 yr in our study, and 4 yr and conserved areas in the study of *Sánchez-Reyes et al. (2021)*). *Kalacska et al. (2004)* also found a lower similarity between the early and late stages of successional stages. However, we found the highest similarity among 10 and 20 yr sites, but *Connell (1978)* attributes this pattern to the possible coexistence of typical species of early, intermediate, and late-successional stages. Thus, our results suggest that the Tamaulipan thornscrub also follows a similar trend to dry forests in tropical regions.

Our third hypothesis said that species with a high ecological importance value in early successional stages will not have high importance values in fragments with vegetation in the late-successional stage. In this sense, we found that Fabaceae species have high representativeness in all stages of abandonment, like recent studies findings at Tamaulipan thornscrub (*Alanís-Rodríguez et al., 2018*; *Pequeño-Ledezma et al., 2018*). It has been documented that Fabaceae species play an important role as pioneer species and can be found in the initial stages of ecological succession due to their capacity to fix atmospheric nitrogen and resistance to drought (*Jia et al., 2020*; *Zhou et al., 2019*). These adaptations could give to this plant family the ability to maintain high abundances in disturbed areas, mainly in the early successional stages. Contrary, in late-successional stages, Fabaceae species are generally less important, because these species require open sites with abundant light.

At the species level, we found that *Acacia farnesiana* was the woody species with the highest abundance and IVI values in the first three successional stages (10, 20, and 30 years of abandonment), but in the >30 yr area *Havardia pallens* and *Diospyros texana* become the species with highest IVI values. These results are similar to the findings reported in several studies from Tamaulipan thornscrub (*Alanís-Rodríguez et al., 2018*; *González-Rodríguez et al., 2010*; *Pequeño-Ledezma et al., 2017*, *2018*), where *A. farnesiana* (along with other native species of Tamaulipan thornscrub) was recorded as species with higher dominance in the four areas of Tamaulipan thornscrub. This could indicate that *A. farnesiana* could play an important role in the early stages of the successional process at Tamaulipan thornscrub vegetation. Thus, *A. farnesiana* could have an essential function in the regeneration of ecosystems that have disturbance such as grazing.

In addition to all aforementioned, we highlight that Tamaulipan thornscrub areas have a relatively fast successional recovery (~30 years). However, *Quesada et al. (2009)* said that regeneration rates in dry forests show relatively lower regeneration rates than we think because in almost secondary forests the coppicing from remainings (stumps and roots) after disturbance theoretically allows these forests to reach maturity faster than the other forests (*Ewel, 1977*; *Murphy & Lugo, 1986*). In the case of this study, the areas have a well-defined history of use for cattle grazing (according to owners), therefore their rapid recovery is basically due to the introduction of pioneer species by some seed vector of surrounding matrix of vegetation. Then, we recommend that future studies also explore

the distance of the seed sources from the surrounding matrix to the studied sites and the intensity of use before abandonment (*Guariguata & Ostertag, 2001*); services of seed vectors, which are variables that have an important role in plant dynamics in degraded sites (*Ramos-Robles et al., 2018*); and changes in the functional characteristics of woody species within fragments (*Guariguata & Ostertag, 2001*). Thus, we highlight the importance of secondary forests for Tamaulipan thornscrub, as they harbor a higher woody plant diversity, which in turn could have the capacity to support the associated fauna that occurs in places with intact vegetation.

## CONCLUSIONS

With the resulting data, we observed the successional trajectory of the Tamaulipan thornscrub fragments. We highlighted that the advance of the successional stages in post-grazing sites of Tamaulipan thornscrub (latest successional stages) generates communities with higher species richness and a higher complex structure than those communities with earliest time of abandonment. The highest similarity among sites with different time of abandonment was observed between sites with closer time of abandonment. On the other hand, a lower similarity was shown between the sites with extremes time of abandonment. *Acacia farnesiana* was the most important and abundant species in the first three successional stages, so we propose it as good pioneer species for restoration programs on Tamaulipan thornscrub ecosystems. We conclude that Tamaulipan thornscrub shows a similar trend of ecological succession to other dry forests and the time of abandonment has a high mediation on plant dynamics in the Tamaulipan thornscrub. Also, we stand out the importance of secondary forests for Tamaulipan thornscrub communities, as they harbor a higher woody plant diversity, which in turn could have the capacity to support the associated fauna that occurs in sites with intact vegetation. Finally, we recommended for future successional studies include aspects of regeneration speed, the proximity of mature vegetation within the matrix where these studied fragments are found, and the interactions of plants with their seed dispersers.

## ACKNOWLEDGEMENTS

We thank R. Domínguez who helped with plant species identification. N. Leal helped with map elaboration.

### Funding

Eduardo Alanis aquired funding from Programa de Mejoramiento al Profesorado (PROMEP/103.5/12/3585). Cristian A. Martínez-Adriano was granted from a CONACYT national postdoctoral fellowship (grants 710775 and 740202). The funders had no role in study design, data collection and analysis, decision to publish, or preparation of the manuscript.

## Grant Disclosures

The following grant information was disclosed by the authors:
Programa de Mejoramiento al Profesorado: PROMEP/103.5/12/3585.
CONACYT national postdoctoral fellowship: 710775 and 740202.

## Competing Interests

The authors declare that they have no competing interests.

## Author Contributions

- Eduardo Alanís-Rodríguez conceived and designed the experiments, performed the experiments, analyzed the data, prepared figures and/or tables, authored or reviewed drafts of the article, and approved the final draft.
- Cristian A Martínez-Adriano analyzed the data, prepared figures and/or tables, authored or reviewed drafts of the article, and approved the final draft.
- Laura Sanchez-Castillo performed the experiments, authored or reviewed drafts of the article, and approved the final draft.
- Ernesto Alonso Rubio-Camacho analyzed the data, prepared figures and/or tables, authored or reviewed drafts of the article, and approved the final draft.
- Alejandro Valdecantos conceived and designed the experiments, analyzed the data, prepared figures and/or tables, authored or reviewed drafts of the article, and approved the final draft.

## Field Study Permissions

The following information was supplied relating to field study approvals (*i.e.*, approving body and any reference numbers):

Permission was granted from landowners was obtained to access study sites. The owners of the properties are not part of any company and their properties are located in rural areas within Linares Municipality.

## Data Availability

The raw datasets are available in the Supplemental Files.

## Supplemental Information

Supplemental information for this article can be found online at http://dx.doi.org/10.7717/peerj.15438#supplemental-information.

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
