# Peer review of "Land abandonment as driver of woody vegetation dynamics in Tamaulipan thornscrub at Northeastern Mexico"

_PeerJ, doi:10.7717/peerj.15438_

## Round 0.1 · original submission · Major Revisions

The scientific article on "Land abandonment mediated the successional changes in woody vegetation communities in Northeastern Mexico" corresponds to the direction of the journal PeerJ. The study of successional processes is an important direction of geoecological research, the purpose of which is to determine the key aspects of the influence of anthropogenic activity on the structure and distribution of vegetation, the change in its biodiversity on different sections of the earth's surface, and the establishment of restoration trends. A special place in the vegetation structure of Northeastern Mexico is played by forests, the preservation and restoration of which in urbanized areas can ensure ecological balance in the region.

This scientific publication is interesting for readers of various profiles and reveals important problems of succession in woody vegetation communities in Northeastern Mexico, which is expressed in changes in horizontal and vertical structure over time. An important development of the authors is the substantiation of the features of the stages of successions in the territories that in the past were subjected to significant anthropogenic influence, in particular - felling, cattle grazing, etc. The structure and nature of the article is satisfactory. However, a detailed study of the publication shows significant shortcomings in the translation. The article's conclusions also require improvement, as their generality and predictability require refinement. In order to improve the quality of the publication, I also recommend taking into account the essential comments from the reviewers, who aim to improve the scientific aspect and highlight the currently relevant problem of tree vegetation succession in Northeastern Mexico.

Reviewer 1 ·

Basic reporting

The article is succinct and informative. It follows the format of PeerJ and includes all suggested sections. Tables and figures are adequate and raw data is provided. The article resorts to the analysis of a chronosequence to assess changes in composition and structure of a subtropical thornscrub community in northeastern Mexico and southern Texas. It is a small article that could benefit from a review of English to improve clarity and readability. The review of literature is not extensive. It includes relevant references to the local vegetation and very general citations about secondary succession but not of similar types of vegetation like tropical dry forests and the analog of Tamaulipan thornscrub on the Pacific Ocean coast: the Sinaloan thornscrub (see for example: Variation in vegetation structure and soil properties related to land use history of old-growth and secondary tropical dry forests in northwestern Mexico. JC Álvarez-Yépiz, et al., Forest Ecology and Management 256 (3), 355-366). Tamaulipan thornscrub vegetation has been little studied in comparison with other vegetation types. This study is not at the forefront of research and lacks originality by posing questions widely documented, particularly that species diversity increases with succession, that beta diversity changes along the extremes of succession, and that the vegetation becomes more complex when comparing pioneer and mature communities. However, this relationship is an important justification for the conservation of mature sections of Tamaulipan scrub vegetation and for protecting the integrity of old remaining stretches of this vegetation.

Experimental design

The experimental design is very simple: four sites with four replicates within sites. There are no measurements of the spatial variance among sites with the same age of abandonment. I understand the technical difficulties, but the article would have greatly benefited from a wider spatial context that included site replicates.

Validity of the findings

The findings, although highly expected, are valid. They corroborate the knowledge that diversity and community structure and composition progress along the secondary succession gradient. Also, provides relevant data about this less studied widely distributed vegetation type.

Additional comments

The text has many small typos that could have been solved before submission. It needs a style and syntax revision. In some cases, the text becomes convoluted and requires rewriting. I added some comments on the attached file. The main criticism of the article is the highly descriptive nature of the data and the way in which it was framed in relation to the ecological literature. The authors decided to discuss the dynamics of succession instead of highlighting the relevance of secondary vegetation and succession in this particular biome. The introduction and the discussion require a more thorough revision and a better framing of the basic questions. In my opinion, the stress on successional events related to the general hypothesis widely resolved by the literature is not the most important contribution of this article but the progression of succession and the way the community fast recovers after disturbance (c. 30 y), particularly after conversion to exotic buffel grasslands. The figures and tables are adequate. The investigation seems performed under appropriate technical and ethical standards and the methods allow for replication only if the original or equivalent sites could be located. The supplementary files are adequate and correct.

Annotated reviews are not available for download in order to protect the identity of reviewers who chose to remain anonymous.

Reviewer 2 ·

Basic reporting

In its current version, the English language of the manuscript (MS) is often unclear and/or awkward, which detracts from the quality and scientific impact of the study. I found the following lines particularly troublesome and hard to comprehend: 1-3, 32-33, 47-50, 52-53, 81-84, 92-93, 100-101, 142, 146-147, 154-155, 240-242, 254-255, 262-266, 280-282, 306-309, 312-314. I strongly recommend that the MS be thoroughly revised by someone who is proficient in English and familiar with the subject matter.

The raw data are supplied and the structure of the MS conforms to PeerJ standards. On the other hand, I found some of the literature cited too narrowly focused on thornscrub succession, even when referring to secondary succession in general (e.g. lines 61-64).

Please find detailed comments on the hypotheses, results, tables and figures below.

Experimental design

This study presents original primary research within the scope of PeerJ. However, the hypotheses presented are actually predictions lacking premises, are awkwardly written and fairly obvious, since most these predictions have been widely corroborated by many previous studies (see for example, Chazdon 2014, Quesada et al. 2009, Guariguata & Ostertag 2001 and references therein). Besides, there seems to be a mistake in the first hypothesis/prediction: in line 104, I suppose you meant sites with the longest (rather than most recent) time of abandonment.

There is a large body of literature on secondary succession (see references above) that is not cited and that contradicts some of the alleged knowledge gaps, such as that “the successional trajectories of plant communities after land abandonment are still poorly understood” (lines 80-81). I therefore fail to see the novelty of this study or its scientific impact and contribution to advancing our knowledge on secondary succession or plant community dynamics.

In terms of the objectives, you state that alpha and beta diversity are explored (lines 96-98) but, as far as I can see, no data on beta diversity is shown in the MS.

The methods are fairly clearly explained, although cluster analysis does NOT evaluate the effect of a given explanatory variable (in this case, successional age) on a given response variable (in this case, similarity in species composition), as implied in lines 148-150. This would require a regression-type analysis instead of an exploratory analysis of classification. Besides, I don’t see the point of running a cluster analysis to classify groups of samples, when there are only four groups to classify.

Finally, the last heading of the Methods section (“Effects of chronosequence on vegetation traits”) makes no sense, since chronosequence is a method commonly used to study succession, not a factor that can affect vegetation traits. The implicit factor behind the chronosequence approach is time after disturbance (i.e. successional age).

References cited:
Chazdon, R. L. Second Growth: The Promise Of Tropical Forest Regeneration In An Age Of Deforestation. (University of Chicago Press, 2014).
Quesada, M., Sanchez-Azofeifa, G. A., Alvarez-Anorve, M., Stoner, K. E., Avila-Cabadilla, L., Calvo-Alvarado, J., ... & Sanchez-Montoya, G. (2009). Succession and management of tropical dry forests in the Americas: Review and new perspectives. Forest Ecology and Management, 258(6), 1014-1024.
Guariguata, M. R., & Ostertag, R. (2001). Neotropical secondary forest succession: changes in structural and functional characteristics. Forest Ecology and Management, 148(1-3), 185-206.

Validity of the findings

The main findings of the study are generally well supported and corroborate what has been previously reported in many studies. However, some of the results are not appropriately or thoroughly presented. For example, Table 1 is never mentioned in the text and does not provide any information that could not be included in Table 2. Indeed, I think Table 2 should include the full scientific names and authorities of the species, their taxonomic families, as well as other relevant information mentioned in the text, such as significant differences in the variables evaluated, which could be indicated with different superscript letters in the last row (totals, not “Add”). Besides, you repeatedly cite Table 2 in the text to illustrate various results that are not shown in this table, such as the effective number of species (lines 195-198), the height profile index (A index, lines 202-204) and wood volume (lines 221-222).

I suggest including a figure or two with results that are mentioned in the text but not shown in Table 2, which could be grouped into species diversity (richness and effective number of species) and vegetation structure (height and/or A index, crown area and/or volume, basal area and/or biomass). By the way, I wonder why you do not show any results on basal area or aboveground biomass, which are highly relevant in terms of vegetation structure and potentially for climate change mitigation.

I found the discussion section rather disappointing. Although you do a good job of comparing your results to those of previous studies in Mexican thornscrub, there are contradictions and inconsistencies in your interpretation of results, which detract from the scientific rigor, quality and impact of the study. For example, the statement that your results “show a progressive recovering trend, where the areas of 30 yr and > 30 yr of abandonment have the closest similarity” (lines 277-279) contradicts your results: “The most similar areas were 10 yr and 20 yr, followed by 20 yr and 30 yr (Table 3)” (lines 229-230). The statement that your results “seem to be a good approach to assessing differences in ecological succession” makes little sense (results are not part of a given methodological approach) and does not logically follow from the fact that you recorded over half of the species reported in previous studies in the studied area (lines 241-244). Also, you need to indicate that your finding that legume species have high representativeness in all stages of abandonment contradicts your third hypothesis (lines 280-283). Moreover, you then present a long argument in support of the dominance of legumes early in succession and their decline later in succession (lines 284-292), without clarifying that, again, this is contrary to the alleged high representativeness of legumes in all successional stages, thus adding to the confusion. I think that the results that you present do not directly address this, so it is hard to gauge whether they support or run contrary to this argument. You could, however, explore if there is a successional trend in the proportion of species that are legumes.

Even more worrisome is your incorrect interpretation of the intermediate disturbance hypothesis and of your own results concerning this hypothesis. As its name implies, this hypothesis posits that diversity peaks at intermediate levels of disturbance, in terms of the frequency and size of disturbance, as well as the time since disturbance. This means that vegetation with no sign of disturbance and no recorded history of disturbance (i.e. old-growth vegetation –labeled as > 30 y-old in this study) should have lower diversity than vegetation with intermediate time since abandonment (e.g. 30 y-old). Therefore, I disagree with your claims that your results on species richness and effective number of species “reinforce the intermediate disturbance hypothesis” (lines 260-262) or “follow [this] principle… since the species composition and abundance is closely related to the advance of successional stages in plant communities” (lines 274-277). You need to clearly establish what constitutes “early”, “intermediate” and “advanced” or “old-growth” stages of the plant communities sampled in this study, and to correctly interpret and cite the classical paper by Connell (1978).

Finally, the main conclusion of the study is poorly stated both in the abstract (lines 52-53) and in the conclusions section (lines 312-314). What do you mean by the time of abandonment “has a high mediation on plant dynamics”? Once more, that successional time strongly influences plant community structure, diversity and composition has been amply corroborated by a vast body of literature and therefore does not really contribute to advance our knowledge of succession or plant community dynamics.

Additional comments

Considering all the major and minor changes that need to be addressed, I don’t think that providing detailed minor comments would be very useful at this stage.

---

## Round 0.2 · accepted · Accept

All reviewers' comments were considered and commented on in the reply letter, the main part of which was taken into account to improve the quality of the manuscript. The English language of the manuscript has been significantly improved, terminological aspects have been clarified, etc. The authors unquestionably revealed the relevance and perspective of scientific research, outlined the main problem areas of scientific work. The manuscript describes well the peculiarity of using the cluster method in scientific research, the effectiveness of which is demonstrated here by the results of the similarity analysis of different sites.
The presented manuscript is qualitatively improved, and the data and actual results obtained are innovative in ecosystem assessment within an ecological succession approach with a well-known abandonment period.